# Solvent-Free and Efficient One-Pot Strategy for Synthesis of the Triazine-Heterocycle Azacyanines

**DOI:** 10.3390/ma15072619

**Published:** 2022-04-02

**Authors:** Xianwu Jiang, Zhuodong Sun, Yu Wang

**Affiliations:** 1School of Chemical Engineering, East China University of Science and Technology, Shanghai 200237, China; jiangxianwu@ecust.edu.cn; 2School of Chemistry and Molecular Engineering, East China Normal University, Shanghai 200241, China; 51210400068@stu.ecnu.edu.cn

**Keywords:** triazine-heterocyclic azacyanine preparation, reaction process optimization, dibromomethane based solution-free method

## Abstract

A novel method with great universality for preparing the electron-rich and electron-deficient triazine-heterocycle azacyanines was presented by using only dibromomethane as a catalysis and solution. The high boiling temperature of dibromomethane has a more flexible reaction condition, allowing all three azacyanine products a chance to yield over 80%. The FT-IR element analysis and all necessary tests, even signal-crystal tests, were executed to firmly confirm that the molecular structure of the azacyanines was accurate. This principal reaction route design that provides a new opportunity for the preparation of azacyanines and their derivatives in a cost-effective and simple process shows great potential for industrial-scale preparation of this important azacyanine intermediate product.

## 1. Introduction

Triazines and their derivatives are important heterocyclic compounds widely used as the intermediate for the synthesis of chemical reagents, commercial medicines, pesticides, and so on [1,2,3,4,5,6,7,8]. With the help of the triazines derivatives, the conjugated azacyanine (N,N’-methylene-2,2′-azapyridocyanine iodide) structure has shown excellent fluorescence property when applied in biological detection, such as in targeting human telomeric DNA by using azacyanine as a detection fluorophore [6] or single-molecule studies [9,10,11]. All these advantages have been widely researched since 1973. This year [12], cyclic azacyanine was successfully synthesized by the reaction of di(2-pyridy1)amine with diiodomethane.

Since 1973, several methods have been developed for the synthesis of cyclic azacyanines. For example, the reflux process was one common method that always used an organic reagent as the solution during the reaction process. For instance, when choosing 2-(2-ethoxyethoxy)ethanol as the solution [13,14], this reaction progress suffered due to its low target product yield value (around 35%) because volatile organic solvents induced complex isolation procedures. It creates a high demand for developing a more efficient and good yield preparation method. The condensation reaction process is another favorable procedure for reaching a high yield of cyclic azacyanine preparation. By using amino-substituted heterocycles, diiodomethane, and an organic solvent as components, the yield of target azacyanine was indeed increased. While the reaction efficiency of this method appears to be extremely susceptible to the electronic environment of the reactant, its amino-substituted heterocycle strongly depends on the electron density of its conjugate structure. The nucleophilicity of the exocyclic amino function may have an unfavorable effect on reaction efficiency. Thus, electron-withdrawing substituents may result in the failure of the reaction and limit its further application. In addition, diiodomethane, used as a catalysis reagent that is feasibly decomposed when exposed to the light and reaction temperature, was also limited by low boiling point values between 8 °C and 69 °C. Furthermore, diiodomethane is not a cost-effective reagent. There is an urgent need to develop a high-yield, un-electron-donating environment and a more efficient procedure [15].

In the present work, diiodomethane is replaced with dibromomethane as a source of methylene (carbine; CH2) to perform the initial bis-quaternization of the aromatic nitrogen atoms in the condensation reaction for the synthesis of triazine-heterocycle azacyanines. The high boiling point of the dibromomethane also gives a more flexible operational condition and is quite suitable as the reactant solution [16,17,18,19]. Thus, a one-pot and high-reaction efficiency strategy for the synthesis of triazine-heterocycle azacyanines with electron-withdrawing substituents was developed. As described in Figure 1, 6H-Dipyri[1, 2-a: 2′, 1′-d][1, 3, 5]Triazin-5-ium Bromide (Azacyanine 1); 3, 9-Dibromo-6H-Dipyri[1, 2-a: 2′, 1′-d][1, 3, 5]Triazin-5-ium Bromide (Azacyanine 2); and 1, 3, 9, 11-Tetrabromo-6H-Dipyri[1, 2-a: 2′, 1′-d][1, 3, 5]Triazin-5-ium Bromide (Azacyanine 3) were successfully prepared, and the yields under optimal conditions reached 90%, 87%, and 80%, respectively. FT-IR, ^1^H NMR, and elemental analysis were used for the characterization of its structure.

The single-crystal analysis of Azacyanine 1 was further executed to firmly confirm that the triazine-heterocycle azacyanine structure was successfully prepared.

## 2. Materials and Methods

### 2.1. Materials

Dibromomethane (99.5%) was purchased from ShanDong Dadi Chemical Co., Ltd. 2-aminopyridine (CP), 2-amino-3,5-dibromopyridine (CP), and ethanol were purchased from Sinopharm Chemical Reagent Co., Ltd. (Shanghai, China). 2-amino-5-bromopyridine (CP) was purchased from Aladdin Reagent (Shanghai, China) Co., Ltd. Propanone (AR) was purchased from Shanghai Reagent Company. If there is no further explanation, reagents were used without any purifications.

### 2.2. Preparation of Azacyanine 1

A mixed solution of 2-aminopyridine (1.0 g, 0.01 mol) and dibromomethane (15 mL, 0.2 mol) was added into a 100 mL flask and stirred in an oil bath heated at 93 °C for 3 h. The reaction was monitored by detecting the pH value fluctuation, which can directly reflect the amounts of the NH_3_ produced during the reaction. Then, the system was naturally cooled, and reaction mixture was collected directly and put into reduced pressure distillation machine to remove excess amounts of dibromomethane [20]. The solid powders were washed with 30 mL propanone for further purifying. The product of 6H-Dipyridino[1,2-a:2′,1′-d][1,3,5]Triazin-5-ium Bromide with yield of 90% was prepared. The detail information of this structure as follows: mp 246–247 °C; IR (KBr, cm^−1^) 3440, 3020, 1500, 1340, 1130, 1000, 901, 820, 700, 665; UV wavelength (nm) 202, 273, 316, 409; ^1^H NMR (500 MHz, DMSO-d_6_):δ6.50 (s, 2H), 7.22 (m, 4H), 8.01 (m, 4H); ^13^C NMR (500 MHz, DMSO-d_6_):δ64.7, 116.8, 121.3, 138.1, 144.0, 152.0; HRMS: *m*/*z* [M-Br^−^] Calcd for C_11_H_10_BrN_3_:184.0869; found: 184.0883. Anal. Calcd for C_11_H_10_BrN_3:_ C, 50.00; H, 3.82; N, 15.91. Found: C, 50.10; H, 3.79; N, 15.71. The name of 6H-Dipyridino [1,2-a:2′,1′-d][1,3,5] Triazin-5-ium Bromide was shortened to Azacyanine 1 if there is no further explanation. Azacyanine 1 (1.5 g) was dissolved into 20 mL of methyl alcohol to form a bright yellow solution in a flask (50 mL) at room temperature. After 2 days, the yellow single-crystals were grown to appropriate size for later analysis by single-crystal X-ray diffraction.

### 2.3. Preparation of Azacyanine 2

The reactant 2-amino-5-bromopyridine with 1.7 g (0.01 mol) was dissolved into dibromomethane (15 mL, 0.2 mol); the reaction was executed in a 100 mL flask and stirred for 4 h in an oil bath heated at 95 °C. The NH_3_ amounts were detected to reflect the reaction progress using the same procedure as described in Section 2.2 of Azacyanine 1 preparation process. The resulting product, 3,9-Dibromo-6H-Dipyridino[1,2-a:2′,1′-d][1,3,5] Triazin-5-ium Bromide, named Azacyanine 2, was prepared. The structure information was as follows: yield 87%; mp 287 °C; IR (KBr, cm^−1^) 3440, 3020, 1490, 1330, 1140, 985, 897, 823, 700, 661; UV wavelength (nm) 211, 261, 305, 406; ^1^H NMR (500 MHz, DMSO-d_6_):*δ*6.33 (s, 2H), 7.21 (d, J = 11.3 Hz, 2H), 8.18 (dd, J = 11.6 Hz, 2H), 8.49 (d, J = 2.75 Hz, 2H); ^13^C NMR (500 MHz, DMSO-d_6_):*δ*65.1, 109.0, 122.9, 138.4, 146.5, 150.9; HRMS: *m*/*z* [M-Br^−^] Calcd for C_11_H_8_Br_3_N_3_: 339.9079; found: 339.9091. Anal. Calcd for C_11_H_8_Br_3_N_3_: C, 31.31; H, 1.91; N, 9.95. Found: C, 31.22; H, 2.08; N, 9.90.

### 2.4. Preparation of Azacyanine 3

A mixture of 2-amino-3,5-dibromopyridine (2.5 g, 0.01 mol) and dibromomethane (15 mL, 0.2 mol) was added into a flask (100 mL) and stirred for 4 h in an oil bath heated at 97 °C. The product, 1, 3, 9, 11-Tetrabromo-6H-Dipyridino[1,2-a:2′,1′-d][1,3,5] Triazin-5-ium Bromide, Azacyanine 3, was collected. The detailed structure information was as follows: yield 80%; mp 289 °C; IR (KBr, cm^−1^) 3440, 3020, 1490, 1330, 1140, 984, 895, 823, 700; UV wavelength (nm) 204, 301, 257, 405; ^1^H NMR (500 MHz, DMSO-d_6_):δ6.36 (s, 2H), 8.61 (d, J = 1.8 Hz, 2H), 8.82 (d, J = 2.75 Hz, 2H); ^13^C NMR (500 MHz, DMSO-d_6_):δ66.1, 108.9, 117.3, 138.3, 148.8, 149.4; HRMS: *m*/*z* [M-Br^−^] Calculated for C_11_H_6_Br_5_N_3_: 495.7290; found: 495.7266. Anal. Calculated for C_11_H_6_Br_5_N_3_: C, 22.76; H, 1.03; N, 7.24. Found: C, 22.81; H, 1.10; N, 7.39.

## 3. Characterization Methods

The FT-IR spectra were tested by using a Nexus 670 spectrometer (Thermo-Nicolet, Thermo Nicolet Corporation, Madison, American). The sample powder was treated through the KBr plate method, with parameters set as 4 cm^−1^ resolution, 32 cycles, and scanning range between 4000 and 400 cm^−1^. UV-Vis spectrum was recorded by a UV 1700 spectrometer; the ethanol solution was used as a reference for baseline correction. The sample with different concentrations was used for the standard curve; scanning range was set as 200–1000 nm with a step of 1 nm. ^1^H and ^13^C NMR spectra were measured by dissolving samples into DMSO-d6 solution and tested by Bruker AVANCE III 400 MHz spectrometer at ambient temperature. The element analysis, including C, H, N, and O samples, was determined by using VARIOEL 3 type element analyzer (Elementar, Langenselbold, German); the accurate contents of each component were calculated by the method reported in the literature [21]. SMART APEX type single-crystal diffractometer (Bruker-AXS) was used for analysis of the single-crystal structure of Azacyanine 1. The 0.3 × 0.24 × 0.14 mm size single-crystal was irradiated by the Mo-ka radiation light. The 2.06° ≤ θ ≤ 27.50° angle range was chosen. The indexation and reduction of diffractive data were executed. SHELEX-97 software was used for resolution of a single-crystal structure.

## 4. Results and Discussion

### Structure Analysis and Thermo-Response Properties of HBPS

The azacyanines were synthesized, as shown in Figure 1. We chose dibromomethane instead of diiodomethane to facilitate the formation of a methylene bridge among different amino groups’ substituted pyridine derivatives and for the synthesis of azacyanine derivatives (Appendix A).

To address this issue, the effect of temperature on the synthesis of azacyanine derivatives was investigated first. The excess amounts of dibromomethane were added to keep the experimental condition reliable, and all other components were added at the same molar ratio, as described in the corresponding experimental parts. All target compounds were successfully synthesized, as confirmed by the corresponding ^1^H NMR spectrum described in the experimental parts. Indeed, dibromomethane is favorable to this reaction, and yields of each reaction were all over 80% at suitable temperatures (Figure 2). More specifically, when the reaction temperature was over the boiling point of the dibromomethane solution, the yield of all three target products improved.

As shown in Figure 2 (black line), the yield of Azacyanine 1 reaches its upper limit value at 110 °C and keeps stable over the later heating process. Similar trends for the yield curves upon heating for Azacyanine 2 and 3 were also observed in Figure 2. Both show an upward trend upon increasing the temperature. The optimized reaction temperature of Azacyanine 2 was 120 °C, and 140 °C for Azacyanine 3. It is clear that the higher reaction temperature was needed to overcome expansion caused by the higher bromo groups’ substituted numbers on the reactant backbone. The electron-withdrawing bromo group reduced the reaction efficiency. As expected, dibromomethane was used as a solvent and catalyzed agent, thus giving the chance to execute the experiment at such a broad temperature range. The higher operation temperature gives this high product yield.

The goal is to develop a more cost-effective azacyanine derivative preparation process. The addition of dibromomethane is a key part of this reaction. The effect of its dosage on the final product yield was also investigated and is shown in Figure 3. All three reactants were executed under 93 °C for 3 h, and like the yield curve upon varying the temperature of Azacyanine 1, increasing dibromomethane amounts increased the speed of the reaction efficiency. Here, dibromomethane is the typical catalysis agent in the first stage (before 15 mL). Later, when it reaches over 15 mL (second stage), the reaction efficiency does not increase as expected. At this stage, the function of dibromomethane is prone to be solvent, reducing the concentration of the components and weakening the reaction efficiency increment. Such a synthesis reaction can be referred to as the solvent-free reaction [17,18,19].

Even though there is no less than one electron-withdrawing substituent on the 2-aminopyridine reactant (used for the synthesis of Azacyanine 2 and 3), dibromomethane still has excellent catalysis properties. The yield obviously increased with an increasing dibromomethane dosage (red and blue curves in Figure 3). Further, at the maximum value of 15 mL dibromomethane dosage, the curve trend was less fluctuated. Dibromomethane is more inclined to the function of the solvent (second stage). As we can see, at this optimized condition, the yields of Azacyanine 2 and 3 were reaching their limitations.

Using this solvent-free, one-pot procedure at this mild condition, heterocyclic azacyanine derivatives were successfully prepared using the dibromomethane and 2-aminopyridine derivatives as reactants, even with the existence of the electron-withdrawing substituents on the aminopyridine backbone. Target products Azacyanine 1, 2, and 3 with yields of 90%, 87%, and 80% were prepared. These yields are much higher than those reported in the literature [13].

Further, the detailed structural information of target products Azacyanine 1, 2, and 3 was investigated. The FT-IR and element analysis results are shown in Figure 4 and Table 1. Figure 4a,c,e shows the FT-IR spectra of aminopyridine derivatives. They were used as reactants to synthesize azacyanines in sequence under the existence of dibromomethane. The FT-IR spectra for their corresponding products, namely Azacyanine 1, 2, and 3, are shown in Figure 4b,d,f. Compared with the reactants spectra, several new peaks occurred, and several disappeared in the spectra of products (Figure 4). Characterized peaks assigned to the aminopyridine derivatives after the reaction are still observed in Figure 4b,d,f, and all the peaks shifted to lower wavenumbers, which were induced by the synthesized conjugated structure from azacyanines derivatives [22].

The sharp band at 3442 cm^−1^ is assigned to ν(N-H) stretching vibrations of the amino group in 2-aminopyridine (Figure 4a), which disappeared in the spectra of Azacyanine 1 (Figure 4b). The bands at 3145 cm^−1^ (νC-H stretching peak) and 1400 and 1380 cm^−1^ (bending vibration peaks of C-H) are characteristic of peaks for the aromatic ring in 2-aminopyridine. All these bands are shifted to a lower wavelength after the synthesis of Azacyanine 1.

Peaks in all three reactant aminopyridine derivatives (Figure 4a,c,e) in the range of 1640~1450 cm^−1^ had the νC=C stretching peaks as well as δN-H formation vibration peaks. There was no δN-H peak observed in the products’ spectra (Figure 4b,d,f), indicating that the conjugated azacyanine structure formed from the aromatic aminopyridine derivatives.

The element analysis results of target azacyanine derivatives are shown in Table 1. Each element proportion for all three target products was perfectly matched with its corresponding theoretical element proportion values (referred to in parentheses in Table 1). The High-Resolution Mass Spectrum (HRMS) results shown in the experiment part, along with all other analysis results shown above, confirmed that heterocyclic azacyanine derivatives were successfully prepared through this one-pot and efficient procedure by using dibromomethane as a catalyst and solvent.

Due to the large conjugated structure of the target azacyanines, they have several strong absorption peaks in the range of 200–600 nm [23]. UV-Vis spectra can be used to reflect the structure difference among different products. Maximum absorption bands of Azacyanine 1 at 210 nm and 240 nm in Figure 5 can be assigned to π→π* electronic transition peaks, while the bands at 281 nm and 415 nm can be assigned to n→π* (marked in Figure 5) electronic transition peaks caused by the formation of the p→π conjugated system between long pair electrons on N, Br atoms, and the C atom on cyclic triazine [24]. As expected, due to the increased amounts of substituted Br atom in triazine of Azacyanine 2 and 3, the substituents have an obvious influence on the n→π* transition bands. The blue peaks shift at 281 nm and 415 nm of Azacyanine 1, at 297 nm and 438 nm of Azacyanine 2, and at 296 nm and 438 nm of Azacyanine 3 (Figure 5). This is because the electron-withdrawing property of the Br group weakens the electron-donating effect of the N-atom on cyclic triazine, leading to the increase in the n-π* energy gap [25]. In addition, it confirms that electron-deficient azacyanines were successfully prepared by an efficient method where the dibromomethane directly reacts with aminopyridine derivatives with electron-withdrawing substituents to form target azacyanine derivatives under the solvent-free condition.

Perspective analysis of the single-crystal can vividly reflect the molecular structure of the target product. Here, we chose a single-crystal test of Azacyanine 1 to obtain more detailed structural information. Its molecular structure was analyzed and derived by using SHELEX-97 software based on the test results from the single-crystal diffractometer, shown in Figure 6 and Table 2. Single-crystal diffraction results of Azacyanine 1 shown in Appendix A and Appendix A represent the unit cell of Azacyanine 1. The main pyridine structure of 2-aminopyridine was the same as the hexatomic ring, which consists of C(1)-C(2)-C(3)-C(4)-C(5)-N(6). Similarly, C(7)-C(8)-C(9)-C(10)-C(11)-N(1) was also derived from the main pyridine structure. The C(5) and chelated Br(1) of the newly formed hexatomic ring in the middle of the Azacyanine 1 is from the reactant dibromomethane, while N(3) belongs to the N atom on 2-aminopyridine.

The bond length results shown in Table 2 are used for obtaining more detailed structural information about the newly formed hexatomic ring located in the middle. The bond length of C(11)-N(3)(1.291(18) Å) is shorter than the length of the single bond C(1)-N(2) (1.369(15) Å), confirming a double bond is formed. While all other chemical bonds located on this hexatomic ring are longer than the bond length of C(1)-N(2), it exhibits single bond characteristics. Its final molecular structure, as shown in Figure 1, also has a strong structure on illustration of the other two azacyanines, since all these products were prepared through a similar procedure. The molecular structure of Azacyanine 1 becomes more accurate and clearer after this single-crystal structure analysis, again confirming that azacyanine was successfully prepared by this novel method.

## 5. Conclusions

By choosing dibromomethane as both a catalyst and solvent, three types of triazine-heterocyclic azacyanines, namely Azacyanine 1, 2, and 3, were successfully prepared, even though the electron-withdrawing substituted aminopyridine was used as a reactant. The reaction procedure was monitored and optimized. Under this optimized condition, the yield of these target compounds reached 90%, 87%, and 80% values, respectively. FT-IR and element analysis were also used to confirm these newly formed cyclic azacyanine products. The existence of the electron-withdrawing substituted group on the azacyanine backbone actually influenced the UV absorption peaks and showed potential to meet the different practical application requirements for the modification of the spectra of azacyanine. To vividly reflect the azacyanine molecular space structure, the single-crystal test of Azacyanine 1 was tested, indicating that azacyanines were prepared. In conclusion, a novel and simple operational method for the preparation of the triazine-heterocycle azacyanines was developed and shown to have a high potential for batch-prepared azacyanines and derivatives on an industrial scale.

## Figures and Tables

**Figure 1 materials-15-02619-f001:**
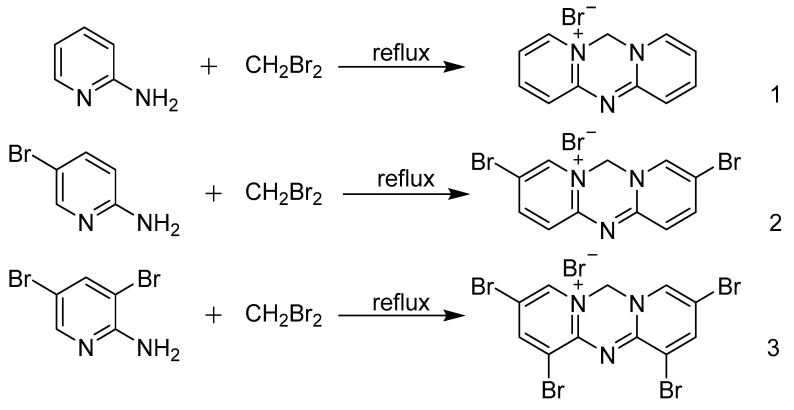
Schematic diagram of reaction between 2-aminopyridine derivatives and dibromomethane; the aminopyridine and various types of bromopyridine derivatives were used as reactants for synthesis of (1) Azacyanine 1, (2) Azacyanine 2, and (3) Azacyanine 3, under the reflux condition with the dibromomethane used as catalysis agent and solution.

**Figure 2 materials-15-02619-f002:**
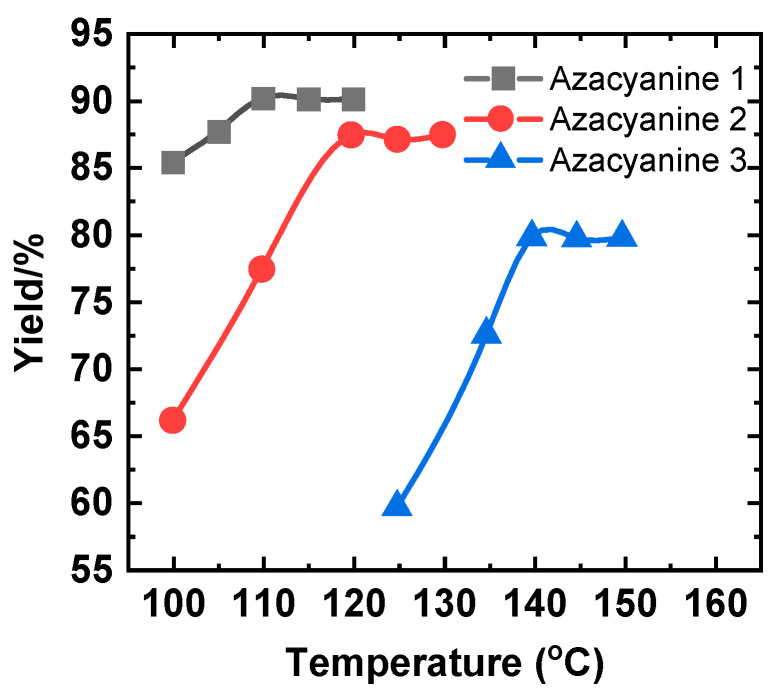
The effect of temperature on the yield of target products: Azacyanine 1 (black), Azacyanine 2 (red), and Azacyanine 3 (blue).

**Figure 3 materials-15-02619-f003:**
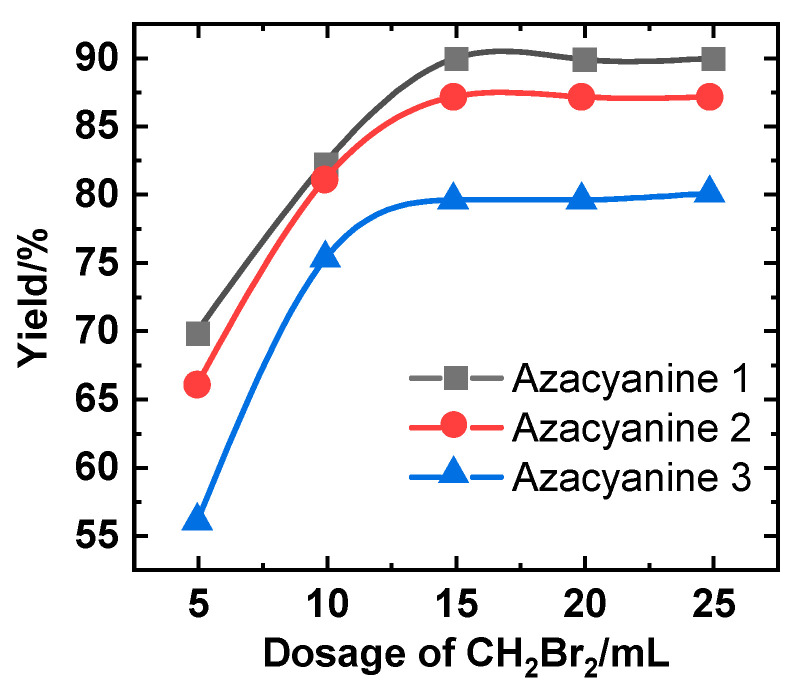
The effect of dibromomethane dosage in the production of azacyanine derivatives: Azacyanine 1 (black), Azacyanine 2 (red), and Azacyanine 3 (blue).

**Figure 4 materials-15-02619-f004:**
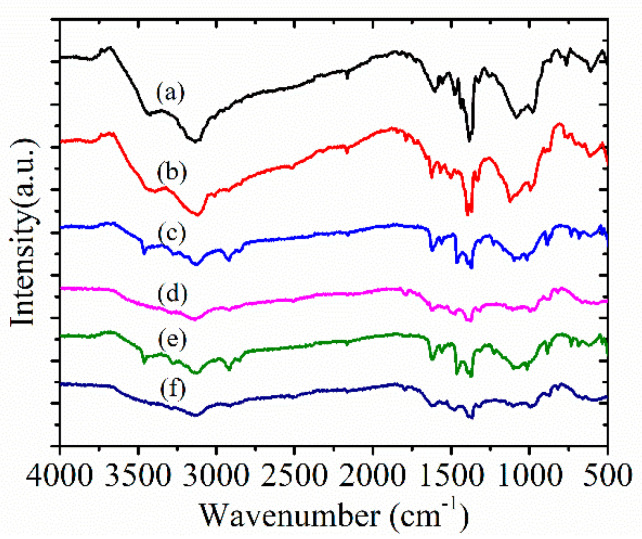
FT-IR spectrum of the reactant aminopyridine derivatives and the target azacyanine products: (a) 2-aminopyridine; (b) Azacyanine 1; (c) 5-Bromo-2-aminopyridine; (d) Azacyanine 2; (e) 4-Amino-3,5-bromopyridine; (f) Azacyanine 3.

**Figure 5 materials-15-02619-f005:**
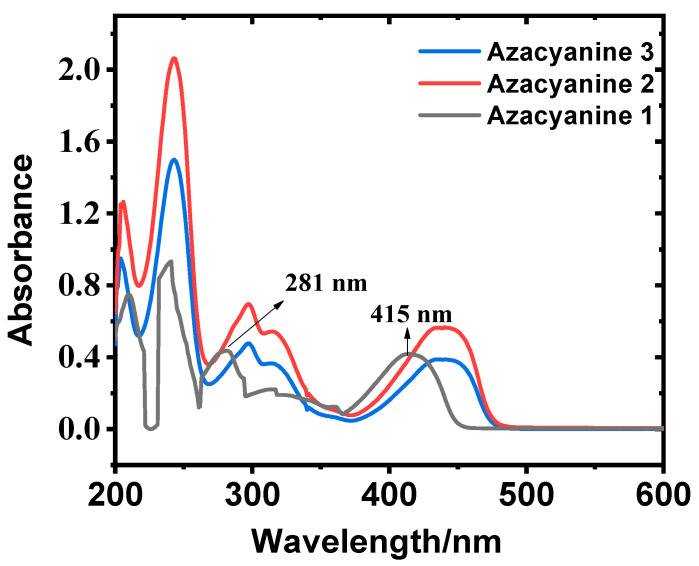
The UV-Vis spectra of the target azacyanine products: Azacyanine 1 (black); Azacyanine 2 (red); Azacyanine 3 (blue).

**Figure 6 materials-15-02619-f006:**
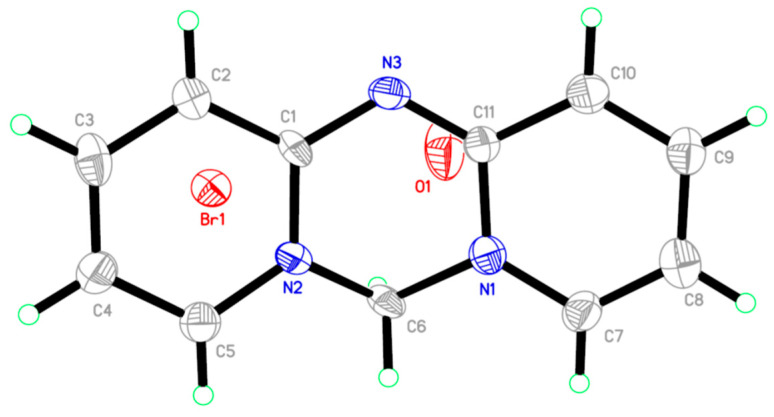
The molecular structure of Azacyanine 1 (the ellipsoids were at a 50% probability level).

**Table 1 materials-15-02619-t001:** The elements analysis results of products: Azacyanine 1, 2, and 3.

Sample ^a^	C (wt%)	H (wt%)	N (wt%)	Br (wt%)
Azacyanine 1	50.10% (50.00%)	3.79% (3.82%)	15.71% (15.91%)	30.37% (30.20%)
Azacyanine 2	31.22% (31.31%)	2.08% (1.91%)	9.90% (9.95%)	57.03% (56.87%)
Azacyanine 3	22.81% (22.76%)	1.10% (1.03%)	7.39% (7.24%)	68.82% (68.97%)

^a^ Theoretical value of each element was used as a reference and shown in parentheses. The standard deviation (SD) of C, H, N, and Br was calculated for each of these three products and is shown here: SD of C is ±0.13; of H, ±0.10; of N, ±0.17; and of Br, ±0.15.

**Table 2 materials-15-02619-t002:** The bond length of Azacyanine 1.

Bond	Bond-Length/Å	Bond	Bond-Length/Å
N(1)-C(7)	1.357(18)	C(5)-H(5)	0.9300
N(1)-C(11)	1.399(18)	C(6)-H(6A)	0.9700
N(1)-C(6)	1.454(17)	C(6)-H(6B)	0.9700
N(2)-C(1)	1.369(15)	C(7)-C(8)	1.34(2)
N(2)-C(5)	1.374(19)	C(7)-H(7)	0.9300
N(2)-C(6)	1.451(17)	C(8)-C(9)	1.40(2)
N(3)-C(11)	1.291(18)	C(8)-H(8)	0.9300
N(3)-C(1)	1.433(17)	C(9)-C(10)	1.38(2)
C(1)-C(2)	1.406(18)	C(9)-H(9)	0.9300
C(2)-C(3)	1.37(2)	C(10)-C(11)	1.42(2)
C(2)-H(2)	0.9300	C(10)-H(10)	0.9300
C(4)-C(5)	1.40(2)		

## Data Availability

All the data are available within the manuscript.

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
