# Peer review of "Solvent-Free and Efficient One-Pot Strategy for Synthesis of the Triazine-Heterocycle Azacyanines"

_materials, 2022, doi:10.3390/ma15072619_

Round 1

Reviewer 1 Report

Wang et al. reported a solvent-free and efficient one-pot strategy for synthesis of the triazine-heterocycle azacyanines. The effect of temperature was investigated. Some analysis of products were reported. However, I put some questions on this work.

1. This strategy was previously studied and the reaction is also simple, so what is the novelty of this work?

2. The procedure requires no metal catalyst, while the reaction is driven by temperature in the reflux condition. Please show the mechanism of the reaction in the presence of dibromomethane as a organocatalyst.

3. The scope of this reaction is very limited. It just focuses on 3 reaction with various substitutions. Please extend your scope. I suggest to carry out the following substitutions: Halide (F, Cl, I), Nitro, Amino, Alkyl, and phenol.

4. Please take the time to revise throughout the manuscript.

5. Please add PDF data of IR and 1H. 13C NMR in the supplementary document for further checking. Thank you!

Author Response

Response 1: The novelty of this work is that choosing dibromomethane reagent as catalysis as well as solution for one step synthesis of heterocycle azacyanine. Compared with others traditional method, like using diiodomethane as catalysis reagent, our method which are shown many advantages, it is more cost-effective, and higher boiling temperature of dibromomethane giving more flexible reaction condition, which have the high potential for synthesis of heterocycle azacyanine with more electro withdraw groups.

Response 2: Thanks for your comment, the mechanism of this reaction was shown in following picture and Figure S1 of revised supplementary document, that the bromo group on dibromomethane was nucleophilic attack on the amino group of pyridine, namely one molar ratio of dibromomethane was reacted with 2 molar ratio aminopyridine through intramolecular cyclization reaction to sequentially synthesis of final structure, triazine-heterocycle azacyanines.

Response 3: Thanks for your comments, our main topic of this manuscript is to confirm that dibromomethane can be used for synthesis of heterocycle azacyanine with at least one type electro-withdrawn group, cause many researches has already been done for synthesis of azacyanines with different substitutions by using dichloromethane as catalysis reagent (Journal of Inclusion Phenomena & Macrocyclic Chemistry, 2006, 54(1-2):129-132; Dalton Transactions, 2009(24):4786; Journal of Organic Chemistry, 2010, 75(12):4292-4295.). We will consider this good suggestion, it is also worth topic for us further investigated.

Response 4: Thanks for the referee’s suggestion. The manuscript were improved and carefully revised.

Response 5: We are grateful to this reviewer for his/her positive comment on our manuscript. The IR results of the structure for target compounds was added into the revised manuscript (Figure 4,Page 6). For the 1H, 13C NMR spectra, here we sent our samples to the cooperative partners, they do only offer stracture information for us.

Please find the detailed responses attached.

Reviewer 2 Report

This manuscript reports about an original low-cost and easy synthesis route for triazine-heterocycle azacyanine compounds and their derivatives, based on the use of dibromomethane as both reaction solvent and catalyst. Owing to the high boiling point of dibromomethane, reaction yields higher than 80% can be obtained. The molecular structure of reaction product was confirmed by FT-IR, elemental analysis, and UV-Vis spectroscopy. The described synthesis scheme is potentially useful for the pharmaceutical industry since conjugated azacyanine structures have shown excellent fluorescence properties in application like telemetric DNA targeting.

This manuscript reports about an original approach for the chemical synthesis of a compound of medical interest and a minimal compound characterization information have been also included. However, clarity is leaching in several points of the manuscript and an overall improvement of the text is required. This manuscript is not adequate for publication on a journal devoted to material science/technology, it must be submitted to an organic chemistry Journal.

Author Response

Response: We are grateful to this reviewer for his/her positive comment on our manuscript. In our study, we have reported a series useful industry intermediate compound, heterocycle azacyanines, it has the highly potential be used in many field, like be used as high-efficiency UV absorber material in detection filed. And our main concern of this manuscript not only focus on synthesis of some new compounds, we actually developed the new procedure for preparation of azacyanines with electro withdrawn substitutions. It should be published in Materials, which at present have the wide group audience choice journal. I hope you and the referees will agree with me that this result is acceptable for publication.

Please find the detailed responses attached.

Reviewer 3 Report

The review can be found in the attached file

Author Response

Response 1: Thanks for reviewer’s suggestions. This items was revised in the manuscript, please find in page 2, line 54, which marked with red colour.

Response 2: Azacyanine 1 (1.5 g) was dissolved in 20 ml of methyl alcohol to form a bright yellow solution in a flask (50 mL) at room temperature. After 2 days, the yellow single crystals were grown in appropriate size for analysis on single crystal X-Ray diffraction. The detailed operational description was added in the experimental part of the revised manuscript. Pleas find in page 2, line 92, which marked with red colour.

Response 3: Thanks for your comments. The CH2Br2 used here is in a liquid state, and we have correctd “dibromomethane solution” instead of the description of “dibromomethane” in Page 3 line 97 of revised manuscript.

Response 4: Figure 2 discussed the effects of different reaction temperatures on the yield of triazine azacyanines both with 15 mL of CH2Br2. Azacyanine 2 and Azacyanine 3 have yield of less than 65% at 93 oC (for azacyanine 2) and less than 55% at 93 oC (for azacyanine 3), but when the temperature reach up to 120 oC and 140 oC, the yield of Azacyanine 2 increases to 87%, and for Azacyanine 3 is 80%. While Figure 3 show dosoge of CH2Br2 at the optimum reaction temperature and having the higher yield value.

Response 5: Thanks for your comments. We have combined all graphs in one Figure 4 of revised manuscript. Considering that the intension of IR spectrum peak of (a) 2-aminopyridine and (b) Azacyanine 1 are stronger than other four results, which the intensity of the absorption peak are similar, so we have separated. It should be combined.

Response 6: Thanks for your comments, we have added the standard deviation of these three batches results in caption of Table 1. And the description of the crystal data was shown as fellow Table, which already been added in the revised supplementary document, the space group only reflect its chiral characteristics, while the bond length descripted in manuscript (Table 2) shows its π-π interaction.

Response 7: Here is the new figure of the unit cell, and been added into the supplementary document of Figure S2.

Response 8: The asymmetric unit of the Azacyanine 1 is hydrate. The source of O atom may be the water in the air absorbed in the process of crystal precipitation or the water contained in the recrystallization solvent (methanol, ethanol). The absorbed water was attached and  combined with Azacyanine 1 to form its crystalline water.

Response 9: Thermal ellipsoids are drawn to the 50% probability level for more detailed reflect its structure characterization, and this description was already added in Figure 6 caption of the revised manuscript.

Response 10: Thanks for your suggestion, the manuscript were improved and carefully revised.

Please find the detailed responses attached.

Round 2

Reviewer 1 Report

The paper can be accepted for publication.

Reviewer 2 Report

I believe the revised manuscript version can be accepted for publication on the journal Materials.

Reviewer 3 Report

I would like to thank the authors for their answers/comments. I believe they addressed everything from my first review and I think the manuscript looks much better now. In my opinion it can now be accepted for publishing in Materials.